# Microbial communities display alternative stable states in a fluctuating environment

**Clare I. Abreu** [1]*, **Vilhelm L. Andersen Woltz** [1], **Jonathan Friedman** [2], **Jeff Gore** [1]*

**1** Physics of Living Systems, Department of Physics, Massachusetts Institute of Technology, Cambridge, Massachusetts, United States of America, **2** Department of Plant Pathology and Microbiology, The Hebrew University of Jerusalem, Rehovot, Israel

☯ These authors contributed equally to this work.
* cabreu@mit.edu (CIA); gore@mit.edu (JG)

**Data Availability Statement:** Access to source data underlying all figures is publicly available at https://github.com/vwoltz-cabreu/alternate_states/blob/master/SourceData.xlsx.

## Abstract

The effect of environmental fluctuations is a major question in ecology. While it is widely accepted that fluctuations and other types of disturbances can increase biodiversity, there are fewer examples of other types of outcomes in a fluctuating environment. Here we explore this question with laboratory microcosms, using cocultures of two bacterial species, *P. putida* and *P. veronii*. At low dilution rates we observe competitive exclusion of *P. veronii*, whereas at high dilution rates we observe competitive exclusion of *P. putida*. When the dilution rate alternates between high and low, we do not observe coexistence between the species, but rather alternative stable states, in which only one species survives and initial species' fractions determine the identity of the surviving species. The Lotka-Volterra model with a fluctuating mortality rate predicts that this outcome is independent of the timing of the fluctuations, and that the time-averaged mortality would also lead to alternative stable states, a prediction that we confirm experimentally. Other pairs of species can coexist in a fluctuating environment, and again consistent with the model we observe coexistence in the time-averaged dilution rate. We find a similar time-averaging result holds in a three-species community, highlighting that simple linear models can in some cases provide powerful insight into how communities will respond to environmental fluctuations.

## Author summary

The effect of environmental fluctuations on community structure and function is a fundamental question in ecology. A significant body of work suggests that fluctuations increase diversity due to a variety of proposed mechanisms. In this study, we compare the effects of constant and fluctuating dilution regimes on simple microbial communities with two or three species. We find that in all cases, the outcome in a fluctuating environment is the same as that in a constant environment in which the fluctuations are time-averaged. This surprising result highlights that in some communities, ecological stable states may be predicted by averaging environmental parameters, rather than by the variation itself.

**Funding:** JG received funding from the Simons Foundation, https://www.simonsfoundation.org/ (grant number 6936338). JG also received funding from the NIH https://www.nih.gov/ (grant number 6936234). The funders had no role in study design, data collection and analysis, decision to publish, or preparation of the manuscript.

**Competing interests:** The authors have declared that no competing interests exist.

## Introduction

In nature, environmental conditions vary over time, and this variation can have significant impacts on the structure and function of ecological communities. Examples of the impacts of environmental variability on community composition include daily cycles of light and temperature that allow nocturnal and diurnal organisms to coexist, and seasonal variation that causes reproducible succession patterns in communities of plants [1], freshwater [2] and marine microbes [3]. Community function can also be strongly influenced by varying environmental conditions. For example, a single rain event can cause up to 10% of annual carbon dioxide production of a forest [4], due in part to enhanced microbial activity in rewetted dry soil [5]. Varying environmental conditions may even cause ecosystems to abruptly and irreversibly change states, such as lakes that shift from clear to turbid due to human-induced eutrophication [6] and reefs that transform from kelp forests to seaweed turfs due to heat waves [7]. Given the inevitability of temporal variability in nature, an improved understanding of how this variability affects ecological communities is essential for understanding natural ecosystems.

Both theoretical [8–13] and empirical [14–18] studies have shown that disturbances can stabilize communities and enhance diversity. For example, temporal fluctuations of light [19] and temperature [20] have been shown to lead to stable coexistence of microbes. There are two types of mechanisms that could cause more species to coexist in a fluctuating environment than in a constant environment: fluctuation-dependent mechanisms and fluctuation-independent mechanisms [21]. Temporal fluctuations have long been proposed as a factor that can increase coexistence to levels not possible in the absence of fluctuations [22–24]. For example, species A may exclude species B in one environment, whereas B excludes A in another environment. Fluctuating between the two environments might lead to coexistence of the two species, despite the fact that they could not coexist in a constant environment. This fluctuation-dependent effect can arise if different species occupy different temporal niches or have relative nonlinearities in their responses to limiting resources [21]. On the other hand, a fluctuation-independent explanation for coexistence in a fluctuating environment is time-averaging: species A and B might coexist when fluctuating between two environments because they coexist in the constant average environment [25]. The former hypothesis predicts that fluctuations can induce coexistence through relative nonlinearities of different species' competitive responses, while the latter hypothesis suggests that coexistence depends only upon the average environment, with or without fluctuations. Choosing between these two hypotheses is in fact a choice between models with linear or nonlinear dependence of per-capita growth on competitive factors [25].

While empirical observations of fluctuation-induced coexistence are common, less attention is focused on how perturbations may lead to outcomes other than increased diversity. For example, alternative stable states have been observed in fluctuating environments, such as in gut microbiota communities subject to laxative treatments [26], intertidal biofilms perturbed by climatic events [27], and regions of forest and barrens disturbed by frequent fires [28]. In these cases, however, it is often difficult to distinguish between alternative states that are simultaneously stable and different states that are stable in different environments. There is some theoretical support for fluctuation-induced alternative stable states in particular systems [29,30], but these predictions are challenging to test in the field due to the difficulty of controlling all ecological factors. Moreover, alternative stable states might appear to be coexistence in a spatially structured environment if isolated patches are occupied by different states; indeed, numerous studies have investigated the hypothesis that environmental heterogeneity increases diversity [31,32]. The proposition that environmental fluctuations may have more complicated

effects on ecosystems than simply affecting diversity is therefore in need of systematic study and demonstration.

In this paper, we make use of highly controllable microbial microcosms to explore the effects of temporal fluctuations on communities. We grow bacterial species in liquid culture with daily dilution, and implement environmental fluctuations by alternating the amount of dilution. The growth-dilution process imposes a tunable death rate on the system, where the dilution factor determines the fraction of cells discarded each day. In a two-species coculture, we observe that fluctuating dilution factors leads to bistability, or two alternative stable states that depend on the initial abundances of each species. We also perform experiments with a constant dilution factor, and we observe bistability at an intermediate dilution factor between the two extremes used in the fluctuating experiments. This fluctuation-independent result suggests that our simple community can be modeled with linear dependence of per-capita growth on competitive factors. We therefore use a simple phenomenological model: the Lotka-Volterra (LV) competition model adapted to incorporate a fluctuating global mortality rate. As a linear model, it predicts that the outcome is independent of the timing of the fluctuations [25], in line with our results. In this model, fluctuating mortality can result not only in bistability, but also in stable coexistence, depending on the strength of interspecies inhibition, which we confirm experimentally. More broadly, the model predicts that an environment with a fluctuating death rate equilibrates to the same outcome as the time-averaged added mortality rate. We test this prediction both in two-species cocultures and in a more complex community with three species, where a fluctuating death rate and a constant death rate lead to the same qualitative outcome. These results suggest that fluctuations can in some cases have easily predictable consequences on community structure.

## Results

To explore the effect of a fluctuating environment on an experimentally tractable microbial community, we performed coculture experiments with fluctuating as well as constant dilution factors. The cultures were allowed to grow for 24 hours and then diluted by transferring a small amount of culture into fresh growth media. Because the total experimental volume remains constant, the amount of culture added from the previous day determines the dilution factor. We began by coculturing slow-growing *Pseudomonas veronii* (*Pv*) and fast-growing *Pseudomonas putida* (*Pp*). (These species' growth rates were measured in monoculture to determine which grew faster (S5 Fig).) At a low dilution factor (10×), *Pv* competitively excludes *Pp*, as the fraction of *Pp* goes to zero from all initial fractions (Fig 1A). At a high dilution factor ($10^4$×), the outcome is reversed, with *Pp* excluding *Pv* as its fraction goes to one from all initial fractions (Fig 1B). We refer to both of these outcomes as competitive exclusion because the final state does not depend on starting conditions; all starting fractions move toward one stable state, which is either zero or one.

Given that there was a single stable state at low dilution and a single stable state at high dilution, we expected that alternating between the two dilution factors would also lead to a single stable state (which could be survival of just one species or possibly stable coexistence of the two). To our surprise, when we performed the coculture experiment alternating between the two dilution factors, we instead observed the emergence of alternative stable states—only a single species survived, but the surviving species depended on the species' initial fractions (Fig 1C). We therefore observe bistability in the fluctuating environment despite the fact that neither environment alone showed bistability.

To explain a possible origin of this emergent bistability, we employed the LV competition model with added mortality, which previously provided powerful insight into how microbial

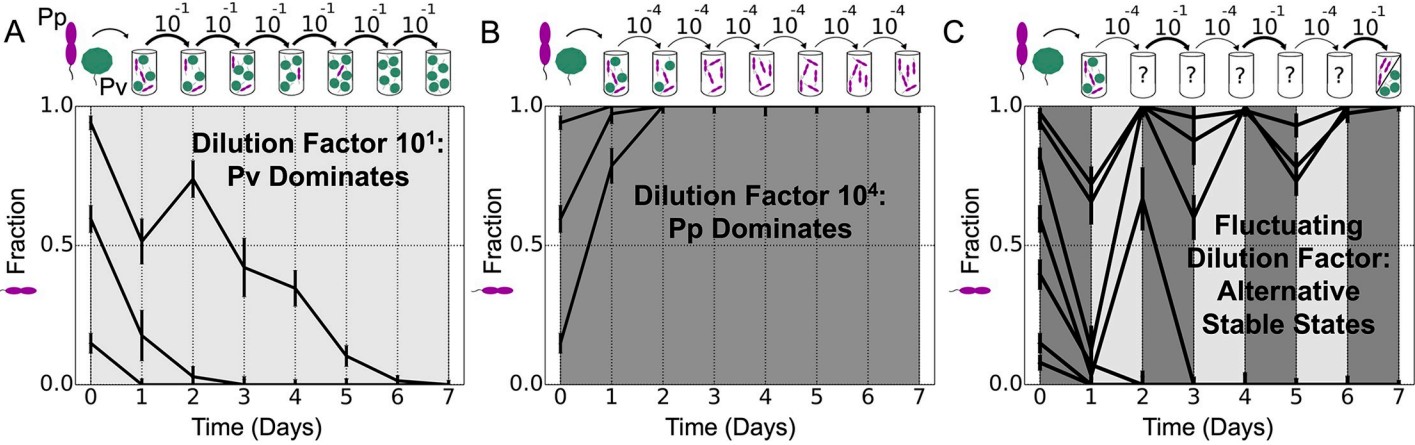

**Fig 1. Experimental observation of alternative stable states in a fluctuating environment. A**: When a coculture of *Pp* (purple) and *Pv* (green) was diluted by a factor of 10 each day (1/10 of the previous day's culture transferred to fresh media, keeping the volume constant), the slow-growing *Pv* dominated, sending the fraction of fast grower *Pp* to zero from several starting fractions. **B**: When the same culture was subject to a much higher dilution factor, $10^4$, fast grower *Pp* dominated. **C**: Fluctuating between the low and high dilution factors shown in **A** and **B** resulted in alternative stable states. Either *Pp* or *Pv* can dominate, depending on their relative initial abundances. In all plots, we qualitatively indicate the dilution factor for that day by the shading of the plot; low dilution factors have a lighter shading, while high dilution factors have a darker shading. Error bars are the SD of the beta distribution with Bayes' prior probability (see Methods).

competitive outcomes shift with dilution rate [33]. The two-species LV model with added mortality is:

$$\frac{\dot{N}_i}{N_i} = r_i(1 - N_i - \alpha_{ij}N_j) - \delta,\qquad(1)$$

where $N_i$ is the abundance of species $i$ normalized by its carrying capacity, $r_i$ is the maximum growth rate for species $i$, $\alpha_{ij}$ is the competition coefficient that determines how strongly species $j$ inhibits species $i$, and $\delta$ is the imposed mortality rate, which is experimentally controlled by the dilution factor, which specifies the fraction of cells discarded per day. In the absence of added mortality $\delta$, the outcome is independent of growth rates $r_i$ and solely determined by whether the competition coefficients $\alpha_{ij}$ are greater or less than one: coexistence and bistability result when both are less than or greater than one, respectively, and dominance results when only one coefficient is greater than one (Fig 2B). The presence of mortality makes the competition coefficients, and thus the outcome, functions of growth and mortality, as can be seen in the reparametrized model (S1 Text):

$$\frac{\dot{\tilde{N}}_i}{\tilde{N}_i} = \tilde{r}_i(1 - \tilde{N}_i - \tilde{\alpha}_{ij}\tilde{N}_j),\qquad(2)$$

where

$$\tilde{\alpha}_{ij} = \alpha_{ij}\left(\frac{1 - \delta/r_j}{1 - \delta/r_i}\right).\qquad(3)$$

If a slow grower dominates at low mortality/dilution, the model predicts that increasing dilution will reverse the outcome and cause the fast grower to win, and at some range of intermediate dilution the pair passes through either a region of bistability or coexistence (Fig 2B). Moreover, a fluctuating dilution rate will lead to the same outcome as the time-averaged rate in the absence of fluctuations. This prediction arises because the per-capita growth rates,

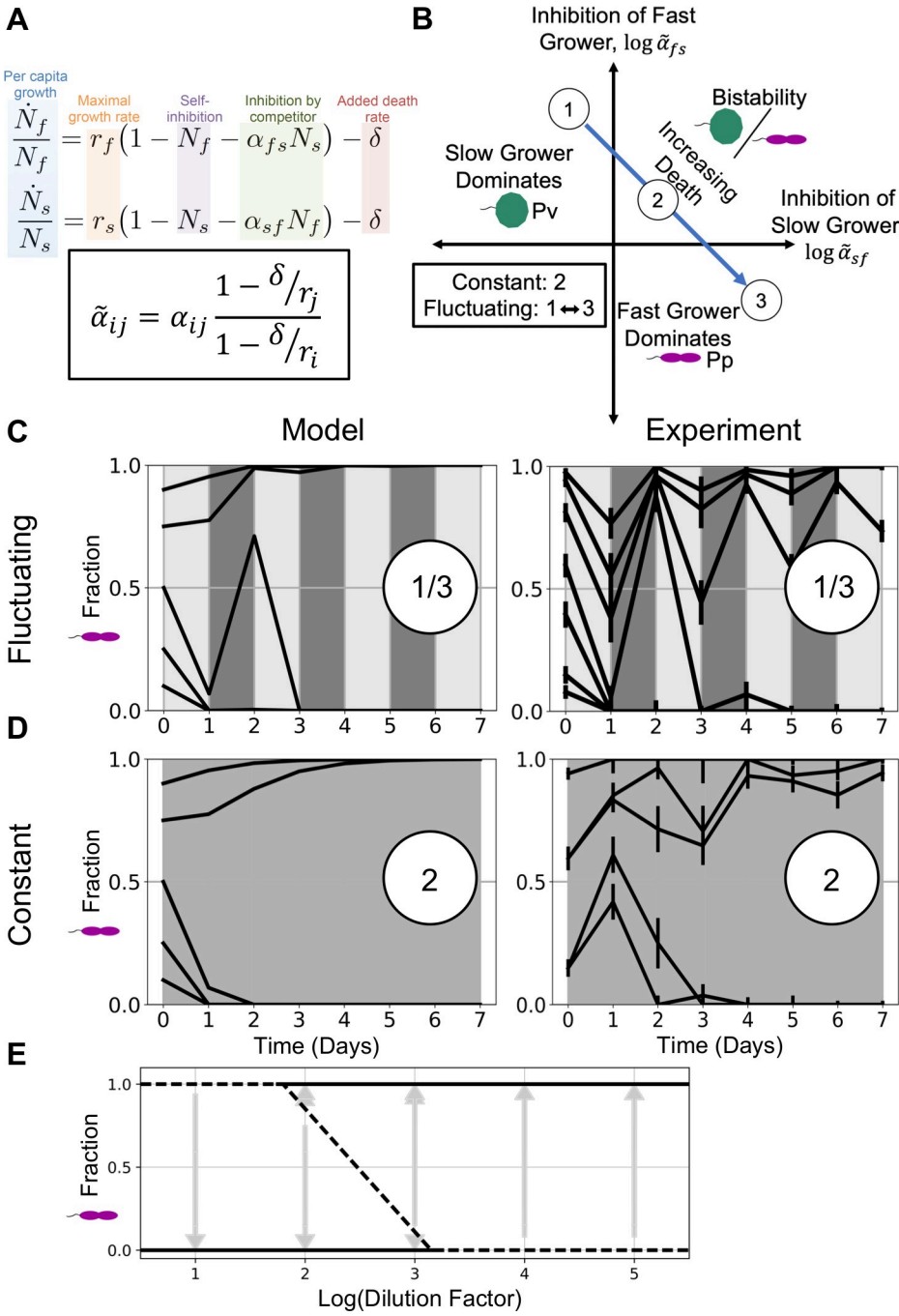

**Fig 2. Bistability occurs in both fluctuating and average environments, confirming model prediction.** To understand the results from Fig 1, we employed a modified Lotka-Volterra (LV) model. **A**: We model daily dilutions by adding a community-wide death rate term $\delta$ to the two-species LV competition model. The per-capita growth rate is a function of a species' maximum growth rate, self-inhibition, and competition with the other species. Because the per-capita growth rate is linear and additive, the model predicts that the outcome of a fluctuating environment should be the same as that of the time-averaged environment (S2 Text). The LV model can be reparametrized to eliminate the death rate $\delta$ by defining new competition coefficients $\tilde{\alpha}_{ij}$, which are functions of death $\delta$ and growth $r$, as shown (S1 Text). **B**: The solutions to the LV model can be represented by a phase space of the re-parameterized competition coefficients $\tilde{\alpha}_{ij}$. If a slow grower dominates at low or no added death, increasing mortality will favor the fast grower, causing the pair to pass through a region of bistability or coexistence on the way to fast grower dominance. Here we have illustrated the trajectory of a bistable pair. **C-D**: To test the prediction that the fluctuating and time-averaged environments are qualitatively equivalent, we cocultured *Pp* and *Pv* at a dilution factor equal to the time-average of the

fluctuating dilution factors in Fig 1, and we observed bistability, confirming this prediction (lower right). The experimental data shown in **C** here is a technical replicate of the same experiment shown in Fig 1C. Additionally, we simulated daily dilutions of the model with both constant (upper left) and fluctuating (lower left) dilution factors and observed good agreement between the model and experimental results. **E**: A bifurcation diagram of the *Pp-Pv* outcomes at all constant dilution factors shows that the fast-growing *Pp* is favored as dilution increases. At each dilution factor, gray arrows represent time trajectories from initial to final fractions; solid lines represent stable equilibria, and dotted lines represent unstable fractions. This diagram was used to estimate the competition coefficients used in simulations (see Methods and S1 Fig). Error bars are the SD of the beta distribution with Bayes' prior probability (see Methods).

$\dot{\tilde{N}}_i/\tilde{N}_i$, in the LV model are linear and additive [25,34], and the steady state reached through a temporally fluctuating mortality $\delta$ is the same state reached by its linear time-average $\langle\delta\rangle$ (S2 Text). The LV model thus makes the simple prediction that an experiment alternating between dilution factors 10 and $10^5$, for example, will end in the same state as one with a constant dilution factor of $10^3$. When averaging dilution factors, we use the geometric mean of the dilution factors because of the logarithmic relation between discrete dilution factor and equivalent continuous rate $\delta$ (S2 Text).

Given our experimental observation of bistability in a fluctuating environment (Fig 1), our next step was to look for bistability in the constant average environment, as the model predicts. We cocultured *Pp* and *Pv* at a range of constant and fluctuating daily dilution factors. The fluctuating dilution factor experiments again led to bistability, replicating our previous results (Fig 2C and 2D). The constant dilution factor experiments revealed a range of fixed points varying with dilution factor: in addition to domination of slow-growing *Pv* at low dilution and that of fast-growing *Pp* at high dilution, we observed bistability at two intermediate dilution factors (Fig 2E). The separatrix, or starting fraction dividing the two stable states, shifts in favor of the fast grower at the higher of these two dilution factors, consistent with the model. These experimental results confirm the model's prediction that the dilution factor can be time-averaged to result in the same qualitative outcome; namely, bistability (see S3 Fig).

As previously mentioned, the LV model also predicts that pairs of species can coexist at intermediate mortality rates. Given this additional prediction, we sought to experimentally verify that coexistence can also result from time-averaging the dilution factors. Based on results of previous cocultured experiments in a similar growth medium [33], we chose another fast grower, *Enterobacter aerogenes* (*Ea*), which is also excluded by the slow-growing *Pv* at low dilution factor. At intermediate dilution factor *Ea* and *Pv* coexist, and *Ea* dominates at high dilution factor (Fig 3D). We then fluctuated between the dilution factors in which either species dominated, as in our initial experiments. This time, we observed stable coexistence in the fluctuating environment (Fig 3B). Furthermore, the stable fraction of *Ea* fluctuated in the neighborhood of the stable fraction of Ea in the constant dilution experiment. In both the bistable pair (Fig 2C and 2D) and the coexisting pair (Fig 3B and 3C), simulations match the experimental trajectories over time, in both constant and fluctuating dilution factors. To confirm that these results could be generalized to more species and pairs, we implemented time-averaging in two additional pairs, one bistable and one coexisting, as well as in other dilution factors with our primary pairs (see S1–S4 Figs). These examples of agreement between the model and experiments emphasize that in our simple bacterial communities, a fluctuating dilution factor may be time-averaged.

While pairwise interactions in a fluctuating environment appear to be well-described by the LV model, the same might not be true of more complex communities. To address this question, we cocultured all three of the previously mentioned species, *Ea*, *Pp*, and *Pv*. Across a range of constant dilution factors, we saw changes in community outcome as a function of the

 

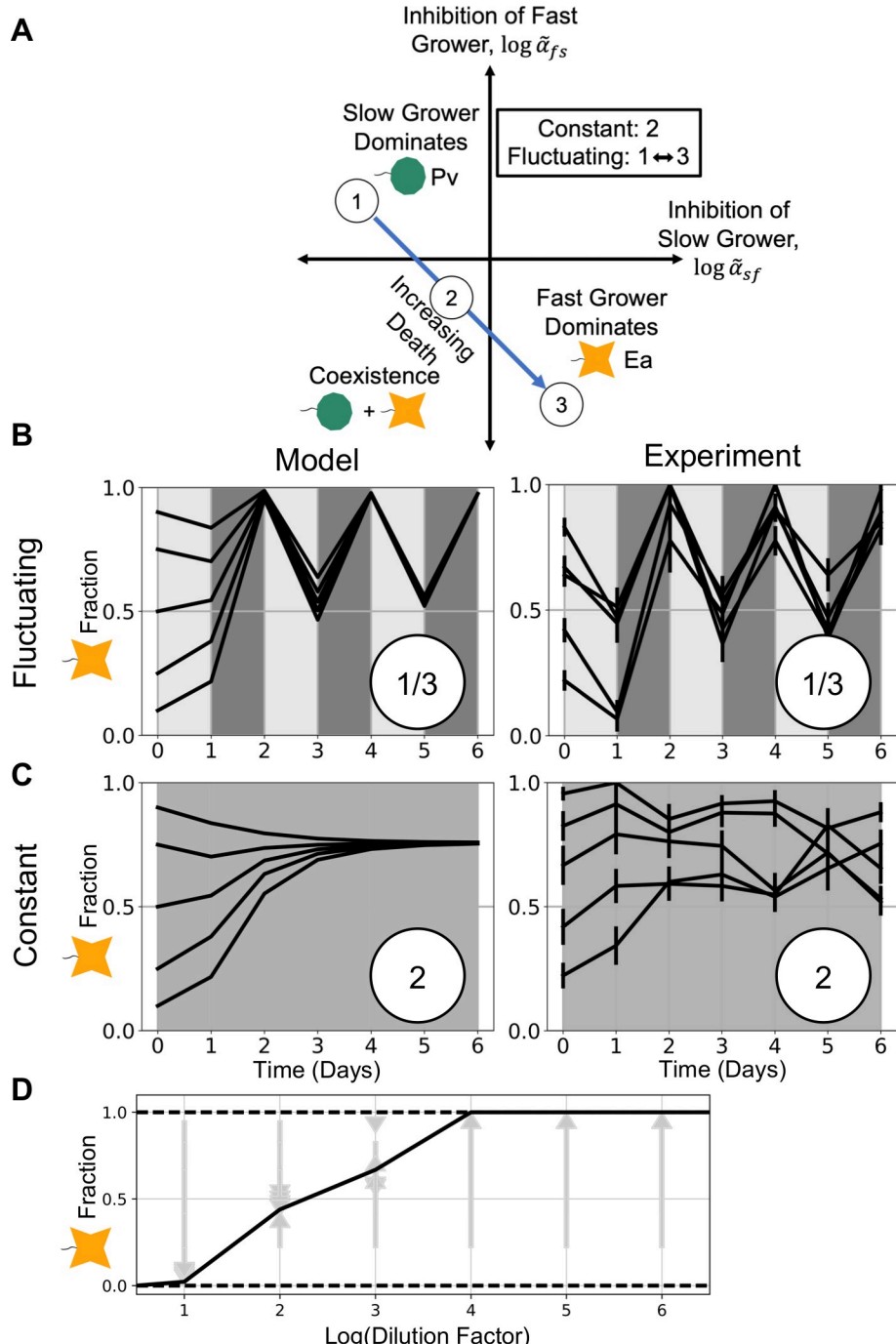

**Fig 3. Coexistence occurs in both fluctuating and average environments, confirming model prediction. A**: As in [Fig 2](), the LV model phase space shows qualitative outcomes divided by the zero-points of the logarithm of the reparametrized competition coefficients. Here we have illustrated the trajectory of a pair that passes through the coexistence region as the death rate increases. **B-C**: When we cocultured slow-growing *Pv* with another fast grower, *Ea*, we observed coexistence in an environment that fluctuated between dilution factors in which either species dominated (upper right). At a dilution factor equal to the mean of fluctuations, we also observed coexistence (lower right), confirming the model's prediction about a time-averaged environment for both types of trajectories across the phase space. Once again, simulations of daily dilutions showed good agreement between the model and experimental results (left). **D**: A diagram of outcomes at all constant dilution factors shows that fast-growing *Ea* is favored as dilution increases. Arrows represent time trajectories from initial to final fractions; solid lines represent stable equilibria, and dotted lines represent unstable fractions. We used this diagram to estimate the competition coefficients used in

simulations (see Methods and S2 Fig). Error bars are the SD of the beta distribution with Bayes' prior probability (see Methods).

dilution factor (S6 Fig), shifting from a *Pv*-dominated state at low dilution to coexistence of *Pp* and *Ea* at high dilution. At an intermediate dilution factor, bistability occurs in the trio between a *Pp-Ea* coexistence state and a *Pv-Ea* coexistence state (Fig 4A). This bistability of coexisting states can be predicted from the corresponding pairwise results, using our previously developed community assembly rules [35]. (It is also worth noting the fact that both coexisting states, *Pp-Ea* and *Pv-Ea*, consist of a member of the *Pseudomonas* genus along with a member of the *Enterobacteriaceae* family. The former prefer to consume organic acids over glucose [36], while the latter prefer glucose [36,37]. Our medium contains both glucose and citric acid, and this compatibility of niche requirements may mitigate competition. We also observed coexistence of two *Pseudomonas* species (S4 Fig), however, which demonstrates that this simple form of niche compatibility is not required for coexistence.) When we fluctuated the dilution factor between the low and high values that average to this intermediate dilution factor, we again observed bistability, with each initial condition ending in the same final state as it did in the constant environment (Fig 4B). These results show that even in a more complex community of three species, a fluctuating mortality rate still leads to the same results as the constant average environment, which is in line with the predictions of the modified LV model.

## Discussion

Regarding theoretical analysis of environmental fluctuations, JW Fox posited in 2013, "The time is ripe for ecologists to . . . embark on a new research program testing the assumptions and predictions of logically valid models of diversity and coexistence in fluctuating environments." [34] Our results highlight a clear experimental demonstration of a simple theoretical prediction about the effects of fluctuations on community structure. Using simple microbial communities with up to three species, we have shown that an experiment with a dilution factor fluctuating between two extremes, both of which typically lead to competitive exclusion, in fact leads to either coexistence or bistability. The outcome of the fluctuating dilution factors is the same as that of the equivalent time-averaged dilution factor, and therefore appears to be independent of fluctuations. Whether the community coexists or forms alternative stable states in the fluctuating environment can be predicted by the community state in a constant environment, providing evidence that a fluctuating environment can be similar to the constant mean environment.

Environmental disturbances have long been thought to weaken competition, thus leading to increased biodiversity [23]. However, both theory [25] and experiments [38] have shown this logic to be incomplete. There are other potential outcomes of fluctuations, such as alternative stable states. In our experiments, we interpreted alternative stable states to be the result of simple time-averaging of dilution factors, meaning that the outcome, whether bistability or coexistence, did not depend on fluctuations per se. Although our results rely on what some may call simplistic predictions, our conclusions are strengthened by the simplicity of both the experimental system and the model. Furthermore, there is some evidence that averaging environmental disturbances may be a feasible strategy in more complex scenarios as well. In one study of multispecies microbial microcosms [39], the multiplicative effect of intensity and frequency of disturbances determined the resulting diversity of a community, indicating that the average overall disturbance was more important than its timing. Other studies [15,40] have found disturbance timing to be the key determinant of diversity within a community, although they did not compare timing to average disturbance. More investigation is needed to

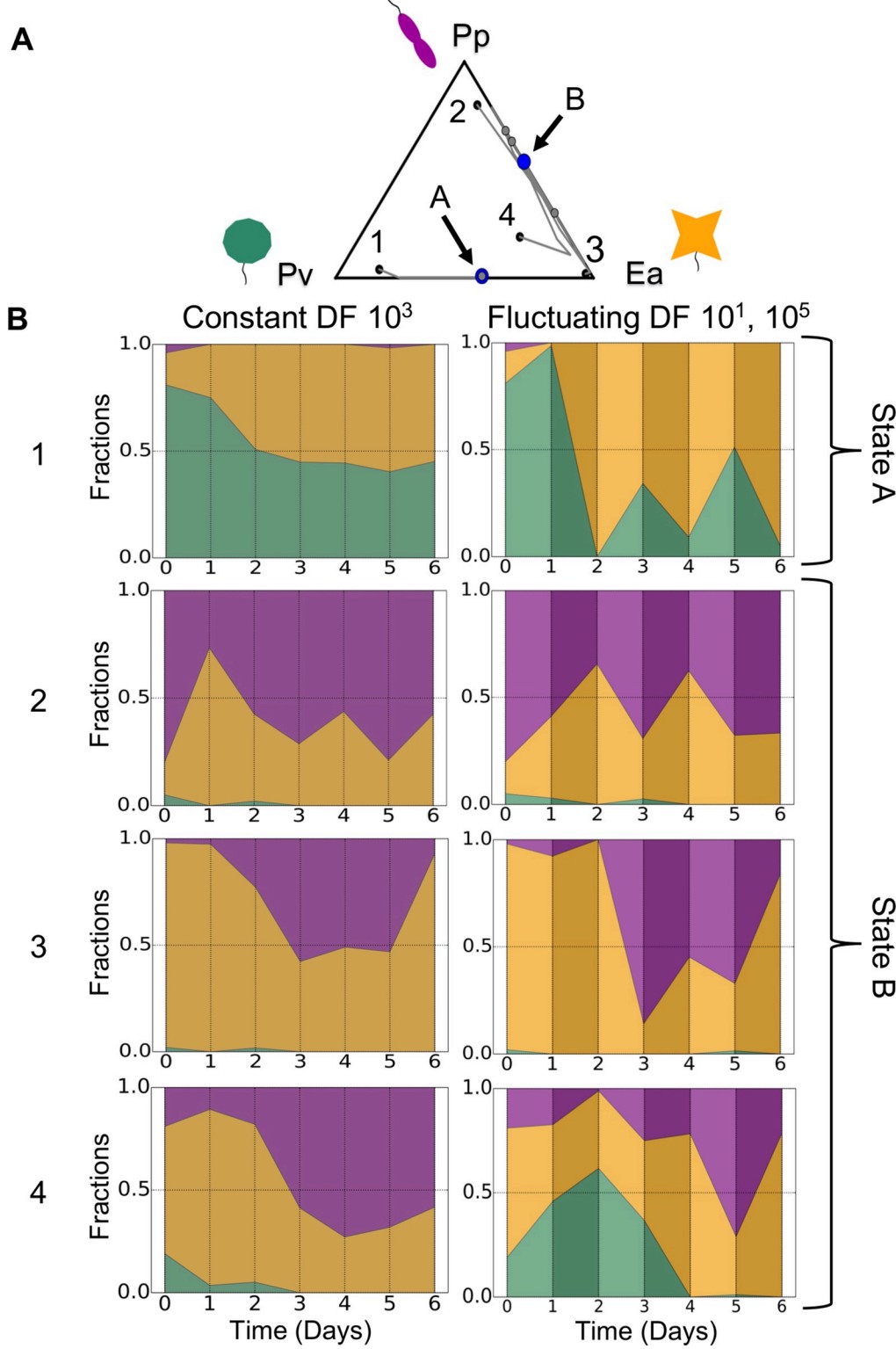

**Fig 4. Fluctuating environment predictably leads to alternative stable states in a three-species community.** To ensure that the time-averaging prediction is not only applicable to simple two-species communities, we tested our results in a three-species environment. **A**: A ternary plot shows the outcomes of a coculture with all three species, *Pv*, *Pp*, and *Ea*, in a constant environment with dilution factor (DF) $10^3$. The time trajectories, indicated by the grey lines, end at one of two alternative stable states (shown in blue; for state B, we plot the average of the three trajectories). In

state A, *Pv* and *Ea* coexist, while *Pp* and *Ea* coexist in state B. **B**: Time series plots show the results of three-species coculture experiments in both constant (DF $10^3$; left column) and fluctuating (between DF $10^1$ and DF $10^5$; right column) environments. As indicated in **A**, three initial fractions end in state B and one ends in state A. We find that the outcomes of a given initial fraction go to the same final state in both the constant and fluctuating environments. This suggests that the ability to time-average the outcomes extends to communities with more than two species (see S6 Fig).

determine whether time-averaging of disturbances is useful in complex yet experimentally tractable communities.

Nevertheless, it is doubtful that all types of disturbances could be time-averaged in even the simplest of communities. In the LV model, mortality/dilution $\delta$ can be time-averaged because it does not covary with other quantities, and the per-capita growth rates are linear and additive [25,34]. Other ecological competition models, such as those that explicitly define resource consumption with the Monod equation, do not predict that the outcome of a fluctuating mortality rate would be the same as that of the equivalent constant environment, because of the nonlinearity of the Monod function with respect to resource concentration [13,41]. A resource-explicit model in which per-capita growth increases linearly with resource concentration [42], on the other hand, would allow for mortality to be time-averaged. While such a model is unrealistic at high resource concentrations, it might represent low-concentration dynamics well. Even in the LV model, however, not all types of fluctuations can be time-averaged; for example, a disturbance affecting the LV competition coefficients $\alpha_{ij}$ would lead to a covariance between the coefficients and species' abundances. As noted previously, temperature fluctuations led to stable coexistence of two species of microalgae [20], and in that case the outcome differed from the outcome at the mean temperature. Indeed, a version of the LV model that assumes temperature acts only upon growth rates $r_i$ does not allow for linear time-averaging of temperature fluctuations [43]. A question for future investigation is whether there are other simple examples of fluctuating parameters that can be time-averaged.

The absence of resources from the LV model is not its only simplification—it also assumes perfect logistic growth and linearity in per-capita effects of species upon each other. While these assumptions are clearly simplifications of the dynamics present within any actual community, the model nonetheless worked well in predicting the outcome of fluctuations in the dilution rate in our microbial community. We used regularly alternating and moderate disturbances, but as the duration and intensity of disturbances increases one expects that eventually the model's predictions will fail due to stochastic extinction caused by finite population sizes. (See S4B Fig for an example of apparent stochastic extinction, in most replicates, when coexistence at a low fraction was expected.) High-frequency fluctuations, on the other hand, may cause qualitatively different dynamics in systems where microbes are able to sense environmental changes [44,45]. Despite the potential complications of modeling environmental fluctuations, the success of the LV model here highlights the relevance of simple, phenomenological models to biological systems.

The effect of environmental disturbances on community structure has become more urgent as many habitats face the effects of climate change. The question of which types of disturbances can be time-averaged is one that may provide insight into the effects of environmental fluctuations on natural communities. Our system was composed of three species of soil bacteria, raising the question of whether communities with more than three species would be similarly affected. A linear time-dependence of interactions and growth rates on added mortality may be more dominant in simple communities, while higher-order effects may become more important in complex communities. If there are nonlinear higher-order interactions in a multispecies community, time-averaging may fail. Additionally, our simple community results do not mean that there were no nonlinearities or covariances in our system, but only that they

were not sufficiently large so as to alter the experimental outcome. Many studies of natural systems have found evidence of nonlinearities and covariances, such as the storage effect [46–48]. In such systems, it may be difficult to disentangle which effects are independent of fluctuations, but our results argue that in some cases, simple linear models retain their predictive power even in fluctuating environments.

## Materials and methods

### Species and media

The soil bacterial species used in this study were *Enterobacter aerogenes* (Ea, ATCC#13048), *Pseudomonas putida* (Pp, ATCC#12633) and *Pseudomonas veronii* (Pv, ATCC#700474). All species were obtained from ATCC. All coculture experiments were done in S medium, supplemented with glucose and ammonium chloride. It contains 100 mM sodium chloride, 5.7 mM dipotassium phosphate, 44.1 mM monopotassium phosphate, 5 mg/L cholesterol, 10 mM potassium citrate pH 6 (1 mM citric acid monohydrate, 10 mM tri-potassium citrate monohydrate), 3 mM calcium chloride, 3 mM magnesium sulfate, and trace metals solution (0.05 mM disodium EDTA, 0.02 mM iron sulfate heptahydrate, 0.01 mM manganese chloride tetrahydrate, 0.01 mM zinc sulfate heptahydrate, 0.01 mM copper sulfate pentahydrate), 0.93 mM ammonium chloride, 1 mM glucose. 1X LB broth was used for initial inoculation of colonies. Plating was done on rectangular Petri dishes containing 45 ml of nutrient agar (nutrient broth (0.3% yeast extract, 0.5% peptone) with 1.5% agar added), onto which diluted 96-well plates were pipetted at 10 μl per well.

### Growth rate measurements

Growth curves were captured by measuring the optical density of monocultures (OD 600 nm) in 15-minute intervals over a period of ~40 hours (S5 Fig). Before these measurements, species were grown in 1X LB broth overnight, and then transferred to the experimental medium for 24 hours. The OD of all species was then equalized. The resulting cultures were diluted into fresh medium at factors of $10^{-8}$ to $10^{-3}$ of the equalized OD. Growth rates were measured by assuming exponential growth to a threshold of OD 0.1, and averaging across many starting densities and replicates (n = 16 for all species).

### Coculture experiments

Frozen stocks of individual species were streaked out on nutrient agar Petri dishes, grown at room temperature for 48 h and then stored at 4˚C for up to two weeks. Before competition experiments, single colonies were picked and each species was grown separately in 50 ml Falcon tubes, first in 5 ml LB broth for 24 h and next in 5 ml of the experimental media for 24 h. During the competition experiments, cultures were grown in 500 μl 96-well plates (BD Biosciences), with each well containing a 200-μl culture. Plates were incubated at 25˚C and shaken at 400 rpm, and were covered with an AeraSeal film (Sigma-Aldrich). For each growth–dilution cycle, the cultures were incubated for 24 h and then serially diluted into fresh growth media. Initial cultures were prepared by equalizing OD to the lowest density measured among competing species, mixing by volume to the desired species composition, and then diluting mixtures by the factor to which they would be diluted daily (except for dilution factor $10^{-6}$, which began at $10^{-5}$ on Day 0, to avoid causing stochastic extinction of any species). Relative abundances were measured by plating on nutrient agar plates. Each culture was diluted in phosphate-buffered saline prior to plating. Multiple replicates were used to ensure that enough colonies could be counted. Colonies were counted after 48 h incubation at room temperature.

The mean number of colonies counted, per plating, per experimental condition, was 49. During competition experiments, we also plated monocultures to determine whether each species could survive each dilution factor in the absences of other species. *Pv* went extinct in the highest two dilution factors ($10^{-5}$ and $10^{-6}$); other species survived all dilution factors.

### Estimating competition coefficients for simulations

In order to simulate the effect of a fluctuating environment on the pairs (Fig 2C and 2D, Fig 3B and 3C), we measured growth rates and carrying capacities (S5 Fig) and estimated the competition coefficients $\alpha_{ij}$. We used the diagrams of pairwise coculture outcomes at all constant dilution factors (Fig 2E, Fig 3D) to estimate the competition coefficients as follows. The different outcomes (dominance/exclusion, coexistence, and bistability) are divided by the zero points of $\log(\tilde{\alpha}_{ij})$ and $\log(\tilde{\alpha}_{ji})$, meaning that the qualitative pairwise outcome changes at a dilution factor where one of the reparamaterized coefficients $\tilde{\alpha}_{ij}$ is equal to one. We used Eq 3 of the main text to solve for the competition coefficient at the boundary dilution factor where unstable (dotted) and stable (solid) lines intersect on the diagrams (Fig 2E, Fig 3D).

### Statistical analysis

The p-values given in S5 and S6 Figs were obtained using two-tailed t-tests. The error bars shown in the time-series plots in Figs 1–3, and S1–S4 Figs are the SD of the beta distribution with Bayes' prior probability:

$$\sigma = \sqrt{\frac{(\alpha+1)(\beta+1)}{(\alpha+\beta+2)^2(\alpha+\beta+3)}}$$

Here, $\alpha$ and $\beta$ are the number of colonies of two different species.

### Supporting information

**S1 Fig. Bistability in pair *Pp-Pv* is reproducible in both constant and fluctuating environmental conditions. A-B**: The data used to generate Fig 2E of the main text shows that *Pp-Pv* is a reproducibly bistable pair at dilution factor (DF) $10^2$, and is sometimes bistable at DF $10^3$, although the lowest starting fraction in Experiment 2 might have been above the separatrix. In both cases, the separatrix changes slightly between experiments. Slow-growing *Pv* is reproducibly dominant at DF $10^1$, and fast grower *Pp* is dominant above DF $10^3$. Two technical replicates (replicates of the experiment from the same colonies) of three starting fractions are shown in **A**, and a second biological replicate (replicates of the experiment from different colonies) of five starting fractions is shown in **B**. **C-E**: Alternative stables states form in a fluctuating environment, and the outcome trends toward that of the average DF ($10^{2.5}$ in panels **C**-**D**, $10^3$ in panel **E**). Note that the starting fractions near the separatrix (~0.6–0.8 in panels **C**-**D**, ~0.1–0.3 in panel **E**) take the longest to equilibrate, and may need longer than seven days to reach an absorbing boundary. Each plot shows one technical replicate of seven starting fractions. Panels **C** and **D** show data sampled from the same biological replicate as panel **A**; panel **E** from the same biological replicate as panel **B**. Error bars are the SD of the beta distribution with Bayes' prior probability (see Methods).
(TIF)

**S2 Fig. Coexistence in pair *Ea-Pv* is reproducible in both constant and fluctuating environmental conditions. A**: The data used to generate Fig 3D of the main text shows a reproducible shift as dilution factor (DF) increases, from dominance of slow grower *Pv* to coexistence to

dominance of fast grower *Ea*. Each plot shows one technical replicate of five initial fractions. **B-D**: Coexistence results in a fluctuating environment if it also results in a constant environment subject to the average DF ($10^3$ in panel **B**, $10^2$ in panel **C**). Furthermore, the coexisting fractions in the fluctuating environments match those of the constant environments. Additionally, competitive exclusion of *Pv* results in a fluctuating environment when it also results in the constant DF ($10^4$, panel **E**). Each plot shows one technical replicate of five initial fractions. Panels **C** and **D** were sampled from a different biological replicate than panels **A** and **B**. Error bars are the SD of the beta distribution with Bayes' prior probability (see Methods). (TIF)

**S3 Fig. Bistability in additional pair (*Pseudomonas citronellolis* (*Pci*)-*Pv*) is reproducible in both constant and fluctuating environmental conditions.** We conducted experiments with another pair found to exhibit alternative stable states, as can be seen from the bifurcation diagram in **A**, which includes data from two different biological replicates. **B**: In an environment fluctuating between DF $10^1$ and DF $10^5$, most trajectories reach the absorbing boundary of zero; the highest initial fraction of *Pci* is very close to the separatrix in the equivalent constant environment (DF $10^3$) and as such does not consistently go to a single final outcome. One should note that these results do not violate the time-averaging prediction of the LV model, since perturbations near a separatrix may cause a trajectory to cross the separatrix and thus take longer to reach equilibrium. Each plot shows one technical replicate of five initial fractions. **C**: The outcome is more predictable when we fluctuate between DF $10^2$ and $10^5$ for a constant equivalent environment of DF $10^{3.5}$, in which the estimated separatrix is about midway between the absorbing boundaries of one and zero. Here we see alternative stable states forming in the fluctuating environment more clearly, depending on whether a starting fraction is closer to one or zero. Each plot shows one technical replicate of five initial fractions, all of which were drawn from a different biological replicate than in panel **B**. Error bars are the SD of the beta distribution with Bayes' prior probability (see Methods). (TIF)

**S4 Fig. Coexistence in additional pair (*Pci*-*Pseudomonas aurantiaca* (*Pa*)) is reproducible in both constant and fluctuating environmental conditions.** We conducted experiments with another pair found to coexist, as can be seen from the bifurcation diagram in **A**, which includes data from two different biological replicates. Error bars are the SEM of all replicates (n = 6; 2 biological replicates of 3 starting fractions each). **B:** In an environment fluctuating between DF $10^2$ and DF $10^6$, for a constant equivalent environment at DF $10^4$, the model predicts coexistence at a stable fraction of *Pci* of ~0.2, as seen in **A**. The failure of coexistence here may be due to stochastic extinction, or domination by *Pa* in the first 24-hour cycle. Neither of these occurrences would violate the time-averaging prediction of the model. Each plot shows one technical replicate of five initial fractions. **C**: In an environment fluctuating between DF $10^3$ and $10^6$ for a constant equivalent environment of $10^{4.5}$, we expect a stable coexisting fraction further from exclusion. As such, we more clearly see the coexistence of the species in this fluctuating environment. Each plot shows one technical replicate of five initial fractions, all of which were drawn from a different biological replicate than in panel **B**. Error bars are the SD of the beta distribution with Bayes' prior probability (see Methods). (TIF)

**S5 Fig. Growth rates for different species were measured using a time-to-threshold method.** To measure growth rates, species were grown in monoculture from a low starting density, with optical density (OD) measured over a period of ~40 hours. Before these measurements, species were grown in 1X LB broth overnight, and then transferred to the experimental

medium for 24 hours. The OD of all species was then equalized. The resulting cultures were diluted into fresh medium at factors of $10^{-7}$ to $10^{-3}$. All of the data used to determine the growth rates is shown in **A**, and one set of growth curves is shown in **B**. Background noise has been subtracted from all curves, and no curves have been smoothed. A threshold OD of 0.1 was chosen, and exponential growth was assumed to occur until this threshold. The time each monoculture took to reach this threshold OD was used along with its initial OD to determine the growth rate. By assuming exponential growth to a threshold, we assume no lag time occurs, but the resulting measurement implicitly incorporates lag: longer lag times will cause the measured growth rate to be lower, while shorter lags will have the opposite effect. **C**: Final growth rate measurements were determined for each species by averaging these measurements across all replicates. **D**: Shown are the measured carrying capacities used for simulating the LV model shown in the figures in the main text. Error bars are the SEM of all replicates (n≥16, per species), of which there were four biological replicates and at least four technical replicates of each (plate reader noise due to factors such as condensation caused some replicates to be excluded for some species).
(TIF)

**S6 Fig. A three-species community reaches similar outcomes in fluctuating and constant environments. A**: Initial and final fractions for each of four starting conditions in each of six constant environments, in which the daily dilution factor (DF) ranged from DF $10^1$ to DF $10^6$. The edges of the ternary plots denote pairwise outcomes; black indicates exclusion, blue coexistence, and red bistability. These outcomes were determined by competing each pair of species in constant environments, the results of which can be seen in the bifurcation diagrams of **C-E**. (Note that one of the pairs, *Ea-Pp*, violates the model's prediction: although *Pp* is the faster grower (see S5 Fig), *Ea* is slightly favored at higher dilution factors. However, *Ea* and *Pp* had the most similar estimated growth rates (p = 0.17, compared to p < 0.01 for the other two pairs of species), which makes the model's prediction more tenuous for this pair.) In **B**, the results of fluctuating the DF in two different regimes are shown, between DF $10^1$ and DF $10^5$, and between DF $10^2$ and DF $10^4$. Both have a constant equivalent DF of $10^3$, and we see that both regimes have the same qualitative outcomes as the constant environment with DF $10^3$, as three initial fractions go to State B and one initial fraction goes to State A, with state labels consistent with Fig 4 in the main text. All plots show data sampled from the same biological replicates.
(TIF)

**S1 Text. Derivation of Lotka-Volterra model modified by added death.**
(DOCX)

**S2 Text. Dilution factors can be time-averaged in the Lotka-Volterra model.**
(DOCX)

## Acknowledgments

We thank the members of the Gore Laboratory for critical discussions and comments on the manuscript.

## Author Contributions

**Conceptualization:** Clare I. Abreu, Vilhelm L. Andersen Woltz, Jonathan Friedman, Jeff Gore.

**Data curation:** Clare I. Abreu, Vilhelm L. Andersen Woltz.

**Formal analysis:** Clare I. Abreu, Vilhelm L. Andersen Woltz, Jeff Gore.

**Investigation:** Clare I. Abreu, Vilhelm L. Andersen Woltz.

**Methodology:** Clare I. Abreu, Vilhelm L. Andersen Woltz, Jonathan Friedman.

**Software:** Vilhelm L. Andersen Woltz.

**Visualization:** Vilhelm L. Andersen Woltz.

**Writing – original draft:** Clare I. Abreu, Vilhelm L. Andersen Woltz.

**Writing – review & editing:** Clare I. Abreu, Vilhelm L. Andersen Woltz, Jonathan Friedman, Jeff Gore.

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
