## [Decision Letter · Decision Letter 0]

14 Feb 2020

Dear Ms. Abreu,

Thank you very much for submitting your manuscript "Microbial communities display alternative stable states in a fluctuating environment" for consideration at PLOS Computational Biology.

As with all papers reviewed by the journal, your manuscript was reviewed by members of the editorial board and by several independent reviewers. In light of the reviews (below this email), we would like to invite the resubmission of a significantly-revised version that takes into account the reviewers' comments.

The reviewers are enthusiastic about your manuscript, and also make several valuable suggestions for edits and changes. Please take account of each of these in considering a revised manuscript.

We cannot make any decision about publication until we have seen the revised manuscript and your response to the reviewers' comments. Your revised manuscript is also likely to be sent to reviewers for further evaluation.

Sincerely,

James O'Dwyer

Associate Editor

PLOS Computational Biology

Stefano Allesina

Deputy Editor

PLOS Computational Biology

The reviewers are enthusiastic about your manuscript, and also make several valuable suggestions for edits and changes. Please take account of each of these in considering a revised manuscript.

Reviewer's Responses to Questions

**Comments to the Authors:**

Reviewer #1: Abreu et al. present an interesting suite of experiments and modeling results, which demonstrate how a fluctuating (density-independent) mortality rate is equivalent to an equivalent time-averaged rate for determining whether a small set of soil-derived bacterial species coexist in co-culture. The experimental methods are straight-forward, as is the LV modeling framework, and I have no technical issues with the manuscript. Overall, I think this is a valuable contribution to the literature. I have a few points that the authors may want to consider in the discussion & interpretation of their results, outlined below.

Specific Comments:

1) In your title, you play up 'alternative stable states', but your model also predicts coexistence (i.e. if the alphas are both negative). Maybe state 'coexistence or alternative stable states'?

2) While your linear model does apply to these 3 strains under the range of fluctuation conditions applied, it is likely that you will find parameters that push your system outside this linear range. Did you try to push your system beyond this range? For example, as you state in the text, large enough variation between dilution rates will result in bottlenecks at the higher magnitude dilutions, which in turn will lead to stochastic extinction events.

3) From a coexistence/diversity perspective, I think you can include some discussion of a spatial interpretation of your alternative stable states as being a form of coexistence. For example, if you have spatial heterogeneity (i.e. spatial patches) then isolated communities that end up in either stable state will show a mixture of species at the 'landscape' scale (i.e. across patches). Lots of good ecology work showing how this type of patch-heterogeneity can promote diversity/coexistence, & prevent extinctions at the landscape scale (but usually in the context of dispersal and neutral patch dynamics -- bistability adds a cool new twist).

4) It's interesting that the pair of species that coexist at intermediate dilution rates are more phylogenetically dissimilar. May be worth discussion how the two Pseudomonads likely overlap more in their niches, compared with E. aerogenes? Indeed, Ea is the coexistence partner for both Pv and Pp in the 3-species co-cultures (i.e. alternate stable states include either Pv + Ea or Pp + Ea). Seems to fit with Ea having compatible niche requirements that mitigate competition? But this might also just be a coincidence.

5) Lines 253-256 -- It might be a bit strong to state that these prior studies 'lack clearly defined predictions and thus make less clear conclusions.' While I agree that your experimental system is quite simple/tractable and the modeling predictions are a bit more clear/cogent, many of these studies do include controlled perturbations in the lab coupled with LV modeling + predictions. I think there are some similarities with these prior studies that further support and complement your conclusions, which could be better leveraged to support your results.

For example, Rodrıguez-Verdugo et al. showed that increasing the frequency of environmental fluctuation resulted in the 2-species community behaving similar to a time-averaged environment, while very low frequencies of fluctuation produced very different results (i.e. one species doesn't have enough time to recover and goes extinct before the environment can flip back to the other condition; Figs. 2-3). This results seems to match quite well with your discussion in the last paragraph if the supplemental text, where you state that "too much time in one dilution factor may move the system to the other side of the separatrix."

Furthermore, in Gibbons et al. the authors show that different intensities and frequencies of disturbance, which ultimately add up to the same time-averaged disturbance rate, have equivalent effects on community diversity (see Fig. 5). In this study, intensity and frequency of two disturbance types (dilution and UV exposure) were modulated independently across (but not within) replicates, unlike your study where the frequency is kept constant for the same community while you modulate dilution intensity within community replicates. In either case, it didn't matter how 'disturbances' are delivered in time, as long as the community is exposed to the same average level of disturbance. Thus, it appears your time-averaging result may be applicable to more complex communities as well.

Finally, I'd like to bring your attention to another paper not cited here by Turkarslan et al. (https://www.embopress.org/doi/full/10.15252/msb.20167058). As you discuss in your manuscript, dilution rate does not appear to influence the alphas (in the LV) or the maximal growth rates. However, other types of disturbances will impact the coefficients in such a model. One example is any fluctuation that forces task-switching (e.g. biphasic growth). Task-switching exacts a metabolic cost in organisms capable of growing under different metabolic regimes. Turkarslan et al. show how higher frequencies can lead to ecological collapse, when organisms are spending all their resources on task-switching without being allowed enough time to grow under either condition (i.e. fluctuation faster than lag phase). Interestingly, they show that if you delete the organisms' ability to sense the environmental change, they stop trying to switch between metabolic modes and collapse is averted. Just one example for how fluctuations that change the niche-landscape of an environment are qualitatively different from fluctuations that do not (e.g. density-independent mortality). Another relevant example would be how dilution/mixing frequency alters oxygen concentrations and therefore changes competition between colony morphotypes of P. fluorescens (e.g. https://onlinelibrary.wiley.com/doi/full/10.1111/j.1558-5646.2009.00758.x). Some discussion of this would be interesting to include, especially in the paragraph where you pose the question about 'which types of disturbances can be time-averaged[?]'.

Reviewer #2: The authors set to test one of the main hypotheses in ecology, namely that environmental fluctuations promote species diversity. They conducted a set of experiments for a simple microbial community and predicted their outcomes with a Lotka-Volterra model with mortality. The authors demonstrated that the outcome of the community dynamics in the environment with a fluctuating dilution rate is the same as in the environment with a time-averaged mortality rate. Moreover, they showed that in different cases fluctuations might lead not only to coexistence but also bistability between two species. In my opinion, the paper outlines a great strategy on how to combine theoretical models and observable experimental data.

I have three main questions, which I think it would be helpful if the authors address when revising their manuscript:

1) It is unclear from the manuscript why the authors observed different outcomes in intermediate dilution rates for two pairs: fast grower Pp and slow grower Pv (bistability), fast grower Ea and slow grower Pv (coexistence). Is it due to the difference in interactions between these two pairs? On lines 140-142, while describing the model, the authors mentioned that “coexistence and bistability result when both alpha_ij are less than or greater than one”, but how can one distinguish between these two? Is there some qualitative understanding of these two cases? I think that it might be a useful discussion as according to the authors these are two different cases - one has increased diversity, another is not.

2) I appreciate that the authors tried to demonstrate that their “time averaging” result holds even for a more complex community of tree species. However, I feel like this example is handpicked, from an earlier Friedman et al. 2017 study, where the authors found that there are no higher-order interactions for Ea-Pp-Pv trio and the outcome of their co-culture can be predicted well using pairwise interaction experiments. I wonder what will happen if there are some non-linear interactions between species (e.g. Pch-Pf-Pv trio from Friedman et al. 2017)? Could a Lotka-Volterra model with some quadratic terms and mortality still be time-averaged? Is there any way to generalize this result to the less specific case?

3) I am a bit confused about the authors mention that “models, such as those that explicitly define resource consumption with the Monod equation … do not make this prediction”. Could the authors elaborate further on why this is the case? Is there an easy way to generalize the result to any other consumer-resource model?

Below are some of my minor concerns on the manuscript:

1) line 30 “cycles of of light”.

2) I suggest not to abbreviate Lotka-Volterra in the manuscript or to do so at the first appearance (line 82).

3) I suggest to introduce Pv as a “slow grower” and Pp as a “fast-grower” somewhere in the text (around line 100-104) as now these definitions appear in the caption of Fig 1.

4) I suggest to highlight lines corresponding to outcomes (Pp or Pv wins) in the same color (e.g. purple for panel B and green for panel A) as species on top in Fig 1 and 2. Then it might be easier to distinguish different trajectories in panel C in case of intersecting lines.

5) I am a bit confused about double grey arrows in Fig 2E and Fig 3D for the intermediate dilution rates. Do they represent different replicates?

6) I would recommend to create a separate legend for “shading” of dilution rates and add it in Fig 2,3,4. Also I think, it might be fine if light grey shade is added on top of Fig 4B to show similar pattern as on previous plots.

7) I feel that it might be beneficial for the reader if authors somehow mark 4 starting points from Fig 4B (initial fractions of bacteria) in Fig 4A. Numbers or different colors probably would work.

8) The authors has demonstrated that for Ea-Pp-Pv trio their modified Lotka-Volterra model is capable of prediction the resulting state. Then, it should be possible to predict a “separator” between 2 attractor states on ternary plot in Fig 4A. Could it be shown on the plot?

9) In Supplementary materials, the authors provided results of dilution experiments for two other pairs of species from the original Friedman et al. 2017 study. However, they are not mentioned anywhere throughout this manuscript and are not needed for the understanding of the main message of the paper. Is there a reason for keeping these?

**Have all data underlying the figures and results presented in the manuscript been provided?**

Reviewer #1: Yes

Reviewer #2: Yes

PLOS authors have the option to publish the peer review history of their article (what does this mean?). If published, this will include your full peer review and any attached files.

Reviewer #1: Yes: Sean Gibbons

Reviewer #2: No
---

## [Decision Letter · Decision Letter 1]

7 May 2020

Dear Ms. Abreu,

We are pleased to inform you that your manuscript 'Microbial communities display alternative stable states in a fluctuating environment' has been provisionally accepted for publication in PLOS Computational Biology.

Best regards,

James O'Dwyer

Associate Editor

PLOS Computational Biology

Stefano Allesina

Deputy Editor

PLOS Computational Biology

Reviewer's Responses to Questions

**Comments to the Authors:**

Reviewer #1: The authors have addressed my comments. I have no other comments or concerns.

Reviewer #2: The authors adequately addressed my comments and improved their manuscript. I feel free to recommend it for publication.

**Have all data underlying the figures and results presented in the manuscript been provided?**

Reviewer #1: Yes

Reviewer #2: Yes

PLOS authors have the option to publish the peer review history of their article (what does this mean?). If published, this will include your full peer review and any attached files.

Reviewer #1: Yes: Sean Gibbons

Reviewer #2: No

---

## [Editor Report · Acceptance letter]

20 May 2020

PCOMPBIOL-D-19-02174R1 

Microbial communities display alternative stable states in a fluctuating environment

Dear Dr Abreu,

I am pleased to inform you that your manuscript has been formally accepted for publication in PLOS Computational Biology. Your manuscript is now with our production department and you will be notified of the publication date in due course.

With kind regards,

Sarah Hammond
